# Inhibition of Phosphoglycerate Dehydrogenase Radiosensitizes Human Colorectal Cancer Cells under Hypoxic Conditions

**DOI:** 10.3390/cancers14205060

**Published:** 2022-10-15

**Authors:** Melissa Van de Gucht, Inès Dufait, Lisa Kerkhove, Cyril Corbet, Sven de Mey, Heng Jiang, Ka Lun Law, Thierry Gevaert, Olivier Feron, Mark De Ridder

**Affiliations:** 1Department of Radiotherapy, Universitair Ziekenhuis Brussel, Vrije Universiteit Brussel, Laarbeeklaan 101, 1090 Brussels, Belgium; 2Pole of Pharmacology and Therapeutics (FATH), Institut de Recherche Expérimentale et Clinique (IREC), UCLouvain, Avenue Mounier 53, 1200 Brussels, Belgium

**Keywords:** serine synthesis pathway, radiosensitivity, hypoxia, reactive oxygen species, colorectal cancer

## Abstract

**Simple Summary:**

Colorectal cancer is the third most prevalent cancer worldwide. Treatment options for these patients consist of surgery combined with chemotherapy and/or radiotherapy. However, a subset of tumors will not respond to therapy or acquire resistance during the course of the treatment, leading to patient relapse. The interplay between reprogramming cancer metabolism and radiotherapy has become an appealing strategy to improve a patient’s outcome. Due to the overexpression of certain enzymes in a variety of cancer types, including colorectal cancer, the serine synthesis pathway has recently become an attractive metabolic target. We demonstrated that by inhibiting the first enzyme of this pathway, namely phosphoglycerate dehydrogenase (PHGDH), tumor cells that are deprived of oxygen (as is generally the case in solid tumors) respond better to radiation, leading to increased tumor cell killing in an experimental model of human colorectal cancer.

**Abstract:**

Augmented de novo serine synthesis activity is increasingly apparent in distinct types of cancers and has mainly sparked interest by investigation of phosphoglycerate dehydrogenase (PHGDH). Overexpression of PHGDH has been associated with higher tumor grade, shorter relapse time and decreased overall survival. It is well known that therapeutic outcomes in cancer patients can be improved by reprogramming metabolic pathways in combination with standard treatment options, for example, radiotherapy. In this study, possible metabolic changes related to radioresponse were explored upon PHGDH inhibition. Additionally, we evaluated whether PHGDH inhibition could improve radioresponse in human colorectal cancer cell lines in both aerobic and radiobiological relevant hypoxic conditions. Dysregulation of reactive oxygen species (ROS) homeostasis and dysfunction in mitochondrial energy metabolism and oxygen consumption rate were indicative of potential radiomodulatory effects. We demonstrated that PHGDH inhibition radiosensitized hypoxic human colorectal cancer cells while leaving intrinsic radiosensitivity unaffected. In a xenograft model, the first hints of additive effects between PHGDH inhibition and radiotherapy were demonstrated. In conclusion, this study is the first to show that modulation of de novo serine biosynthesis enhances radioresponse in hypoxic colorectal cancer cells, mainly mediated by increased levels of intracellular ROS.

## 1. Introduction

Colorectal cancer (CRC) is the third most prevalent cancer worldwide and accounts for about 10% of cancer-associated mortality in the Western world. Despite a declining mortality rate due to effective cancer screening measures, an important increased incidence is expected in the coming years due to population ageing, poor dietary habits, smoking, sedentary lifestyle and obesity in developed countries [1]. To date, the standard of care for CRC patients consists of surgical resection combined with chemotherapy and/or radiotherapy (RT) in the case of advanced CRC. Despite significant advances in treatment options, patients with radioresistant tumors often relapse.

In recent years, an augmented de novo serine synthesis pathway (SSP) [2] has been revealed in distinct types of cancers, including melanoma [3], breast [4], lung cancer [5] and multiple myeloma [6]. The importance of the SSP is increasingly apparent due to its proven contributions in tumor initiation, tumor progression and therapy resistance [7,8]. In short, the SSP consists of three enzymatic reactions catalyzed by phosphoglycerate dehydrogenase (PHGDH), phosphoserine aminotransferase (PSAT1) and phosphoserine phosphatase (PSPH), resulting in the generation of serine. PHGDH is the first and rate-limiting enzyme of the SSP, responsible for shunting approximately 10% of glucose from the glycolysis. Overexpression of PHGDH was first described in triple-negative breast cancer (TNBC) and has drawn considerable attention in the field of cancer research ever since [4,9,10].

The interplay between reprogramming metabolic pathways and radiation is an appealing strategy to improve therapeutic outcome. Serine is classified as a nonessential amino acid; however, metabolically, serine plays an essential role in multiple cellular processes [2]. Intriguingly, serine is able to perturb redox homeostasis and regulate the production of glutathione (GSH) and its cofactor nicotinamide adenine dinucleotide phosphate (NADPH), which act as the main gatekeepers of reactive oxygen species (ROS) homeostasis [11,12]. The latter is dependent on the balance between oxidants and antioxidants. Cancer cells intrinsically possess a disturbed redox balance, and as a consequence are often unable to cope with perturbed ROS levels. Furthermore, ROS are the main effector molecules of radiation and are responsible for about two-thirds of radiation-induced DNA damage [13]. Depletion of these main antioxidants can strongly potentiate radioresponse, with several compounds currently under (pre-)clinical investigation [14,15,16,17,18].

Furthermore, the SSP accounts for approximately 50% of total anaplerotic flux of glutamate into the tricarboxylic acid (TCA) cycle [10]. PSAT1 is responsible for the conversion of glutamate into alpha-ketoglutarate (aKG) [19], thereby regulating the production of the latter [7]. In cancer cells, glycolysis and TCA cycle intermediates are deviated in order to produce essential precursors to support growth and proliferation instead of sustaining bioenergetic production [12]. In the case of TCA intermediate impairment, alteration of mitochondrial respiration and permuted oxygen consumption will take place. Altered mitochondrial respiration through oxidative phosphorylation (OXPHOS) as well as impaired oxygen consumption are both interconnected with radioresponse. Indeed, this impairment results in higher oxygen levels in the tumor microenvironment, thereby alleviating hypoxic tension, which is the major cause of radiotherapeutic resistance.

Intriguingly, augmented de novo SSP activity and its prognostic significance in CRC have only been highlighted in several publications [11,20,21,22,23,24]. It has already been demonstrated that certain CRC cell lines exert a phenotype that is (partly) dependent on serine for their growth/metabolism [25,26]. Additionally, CRC is an ideal model to study potential radiosensitizers [27,28], as radiotherapy is given in a preoperative setting to advanced CRC patients within their standard-of-care treatment regimen. Indeed, combination therapy with compounds that radiosensitize tumor cells is of considerably higher value when the gross tumor volume is still present at the time of irradiation, as compared to postoperative radiotherapy. The only association between radiotherapy responses and the SSP has been observed in head and squamous cell carcinoma cell lines, where radioresistance was linked with an increased metabolic profile of serine and glycine [29]. To the best of our knowledge, no research has been performed investigating the influence of SSP modulation on radioresponse to date. This study aimed to examine possible metabolic alterations related to radioresponse upon PHGDH inhibition and the consequent radiosensitizing effect in human CRC cells.

## 2. Materials and Methods

### 2.1. Xena and TCGA Colorectal Cancer Cohort Analysis

Patient-derived mRNA expression profiles (RNAseq–RSEM norm.count) were queried from the Xena Explorer website (https://xenabrowser.net (accessed on 4 November 2020)). Next, expression levels for the different SSP enzymes were collected in healthy and CRC tissues using the Xena Explorer tool. Expression profiles of GTEx normal tissue were compared with expression profiles of CRC patients obtained from TCGA. A cohort of 639 matching samples with 331 samples in TCGA dataset and 308 samples in the GTEx dataset were obtained for PHGDH, PSAT1 and PSPH. Overall survival was assessed for PHGDH, PSAT1 and PSPH in rectum adenocarcinoma using the Pan-cancer database from TCGA. The Kaplan–Meier Plotter tool (https://kmplot.com (accessed on 6 September 2022)) was used to generate the survival curves [30]. Next, we queried mRNA profiles (Log RNA seq V2 RSEM) from the cBioPortal website in form of z-score-transformed data and clinical data from CRC patients [31,32]. The Buffa, Winter and Ragnum hypoxia score analysis of the samples where PHGDH has a z-score higher than 1.5 was performed directly on the cBioportal website (cut-off between high and low PHGDH, z-score of > 1.5).

### 2.2. Cell Lines and Chemicals

Human HCT116 and DLD-1 CRC cell lines were obtained from American Type Culture Collection (ATCC, Manassas, VA, USA). All experiments were conducted in Roswell Park Memorial Institute (RPMI) 1640 (Thermo Fisher, Brussels, Belgium), or for experiments in the absence of extracellular serine, with Minimal Essential Media (MEM) (Thermo Fisher, Brussels, Belgium) supplemented with 10% bovine calf serum (Greiner Bio-One, Vilvoorde, Belgium). Chemical reagents were obtained from Sigma–Aldrich (Antwerp, Belgium) unless otherwise stated.

### 2.3. Treatments

Cells were grown to subconfluence and exposed to the PHGDH inhibitor NCT-503 [33] for 5 h unless for cytotoxicity and radiation experiments, where cells were exposed to NCT-503 for 16 h at indicated concentrations. ROS scavenger N-acetyl cysteine (NAC) was added at 10 mM to cultures both 1 h prior to and during treatment with NCT-503.

For the in vivo treatment, NCT-503 was first dissolved in ethanol (100%). Afterwards, polyethylene glycol 300 (PEG300) (35%) and hydroxypropyl-beta-cyclodextrin (30% saline solution) (60%) were added. The vehicle was composed of 5% ethanol, 35% PEG300 and a 60% hydroxypropyl-beta-cyclodextrin saline solution in the same ratios as described above.

### 2.4. MTT Assay

Cytotoxicity of NCT-503 was assessed by MTT assay as described elsewhere [16]. Absorbance was measured at a wavelength of 540 nm by using a spectrophotometer (Bio-Rad, Hercules, CA, USA). Cell viability was determined by dividing the absorbance values of treated cells to those of untreated cells.

### 2.5. Apoptosis Assay

Apoptosis was analyzed by flow cytometry using double staining with lipophilic Annexin V and 7-amino actinomycin D (7-AAD) (Abcam, Cambridge, UK), as described elsewhere [16]. Apoptosis was determined by flow cytometry (BD LSR Fortessa, BD Bioscience, Franklin Lakes, NJ, USA) and analyzed by Flowjo7.6 (Treestar Inc., Woodburn, OR, USA).

### 2.6. Radiation and Clonogenic Assays

In order to induce hypoxia (0.1% O_2_), treated cultures were placed into a hypoxia chamber. Subsequently, vacuum was induced and 94% N_2_, 5% CO_2_ and 1% O_2_ were infused. This cycle was repeated 4 times. A total of 0.1% O_2_ was thereafter obtained by calculating the volume of atmospheric air oxygen needed and injecting this into the hypoxia chamber. After overnight treatment, hypoxia chambers and 24-well plates (normoxia) were irradiated at indicated doses with 6MV Linac Varian truebeam stx (Palo Alto, CA, USA, BrainLab AG). Cells were reseeded for colony formation as described elsewhere [25]. After 10-12 days, cultures were fixed with crystal violet and colonies (>50 cells) were counted. Survival curves were fitted to the linear quadratic model using GraphPad Prism 9 software (GraphPad Prism Software Inc., San Diego, CA, USA). Radiosensitization was expressed as an enhancement ratio (ER) determined at a survival fraction of 10-1.

### 2.7. Western Blotting

Western blot analyses were performed as described elsewhere [18]. Briefly, cells were lysed in 1% triton-X buffer supplemented with phosphatase inhibitor (P5726), protease inhibitor (P8340) and leupeptin trifluoroacetate (L2023). Lysates were centrifuged, and protein concentration was determined using the Bio-Rad DC protein assay (Bio-Rad 500-0116). Equivalent amounts of proteins were loaded on a 12% resolving acrylamide gel. Protein transfer was conducted overnight at 4 °C using a nitrocellulose membrane [0.45 µm, Thermo 88018, Thermo Fisher, Brussels, Belgium). Afterwards, membranes were blocked with 5% BSA in TBS and washed (TBST). Blocked membranes were labeled with primary antibodies overnight at 4 °C. Primary antibodies were labeled with near-infrared secondary antibodies (IRDyes 680 RD or 800 CW, LI-COR Biosciences, Lincoln, NE, USA), detected and quantified using Odyssey Fc Imaging System (LI-COR Biosciences, Bad Homburg, Germany). Primary antibodies were anti-beta ACTIN (A1978), anti-PHGDH (NBP1-87311), anti-PSAT1 (NBP1-32920) and anti-PSPH (NBP1-56848).

### 2.8. Serine Measurement

Measurement of total serine concentration and assessment of D- versus L-enantiomers were conducted using a fluorometric DL-serine assay kit (BioVision—Gentaur, Kampenhout, Belgium) according to the manufacturer’s instructions. Briefly, cells were treated with NCT-503, trypsinized and lysed by sonification for 1 min, centrifugated at 10,000 rpm for 15 min at 4 °C and ultimately deproteinized by using 10kDa spin columns. Cells were assayed in triplicate. Results were read in a fluorescent manner by Glomax plate reader (Promega, Madison, WI, USA).

### 2.9. ROS Levels

Intracellular levels of ROS were detected by flow cytometry using an oxidation sensitive fluorescent probe, namely 5-[6]-chloromethyl-2′,7′-dichlorodihydro-fluorescein diacetate (CM-H2DCFDA) (Abcam, Cambridge, UK), as described elsewhere [16]. Briefly, cells were treated with NCT-503 and stained with 5 µM CM-H2DCFDA at 37 °C for 30 min and analyzed by flow cytometry.

### 2.10. NADPH Level

Intracellular NADPH levels were measured with a commercial kit (AAT Bioquest, Sunnyvale, CA, USA) according to the manufacturer’s instructions. Briefly, cells were treated with NCT-503 and washed in serum-free medium. Cells were prepared in 0.5 mL serum-free medium and incubated with 1 µL of JZL1707 NAD(P)H sensor at 37 °C for 60 min. Afterwards, cells were washed and kept in assay buffer until analysis by flow cytometry.

### 2.11. Glutathione Assay

GSH levels were measured with a commercial GSH assay kit (Sanbio, Uden, The Netherlands) as described elsewhere [16]. Briefly, cells were treated with NCT-503, washed twice with PBS and resuspended in cold MES buffer. Hereafter, cells were lysed by sonification for 1 min, centrifugated at 10,000 rpm for 15 min at 4 °C and ultimately deproteinized by using 10 kDa spin columns. Then, 50 µL of the collected supernatant was added to 150 µL assay cocktail, and absorbance was measured at 410 nm using a spectrophotometer over 30 min with five-minute intervals.

### 2.12. Alpha-Ketoglutarate Production

aKG levels in cultured cells were measured using a fluorometric assay kit (BioVision—Gentaur, Kampenhout, Belgium) following the procedure recommended by the manufacturer. Briefly, cells were treated with NCT-503, trypsinized and lysed by sonification for 1 min, centrifugated at 10,000 rpm for 15 min at 4 °C and ultimately deproteinized by use of 10 kDa spin columns. Cells were assayed in duplicate. Results were read by a Glomax plate reader.

### 2.13. Metabolic Profiling

Oxygen consumption rate (OCR) and extracellular acidification rate (ECAR) were measured using a Seahorse XF96 plate reader (Agilent Technologies, Santa Clara, CA, USA). Briefly, 2 × 10^5^ cells were seeded in a 96-well plate and treated with NCT-503 overnight. Cells were then equilibrated in unbuffered Dulbecco’s Modified Eagle Medium (DMEM) medium with 2 mM glutamine and 10 mM glucose at 37 °C in a CO_2_-free incubator and then measured using a Seahorse analyzer. In order to obtain detailed information about mitochondrial respiration, specific inhibitors consisting of oligomycin (1 µM), FCCP (1 µM), rotenone (0.5 µM) and antimycin A (0.5 µM) were added sequentially.

### 2.14. In Vivo Tumor Xenograft Experiments

Six week-old female athymic nude mice were purchased from Charles River. All experiments received the approval of the Ethics Committee from the Vrije Universiteit Brussels (VUB) (No. 20-552-1) and were carried out according to national care regulations. HCT116 cells (4 × 10^6^) were subcutaneously inoculated into the left flank of mice. When the tumors reached 150 mm^3^, mice were treated for 4 days with 40 mg/kg through intraperitoneal injection (100 µL i.p.) and/or radiation (3 daily fractions of 5 Gy), starting after the second administration of NCT-503, by using a 6 MV linac (Elekta, Stockholm, Sweden). The day prior to radiation, dedicated computed tomography (CT) with fixed kV and 3 mm slice thickness was performed for each mouse separately for lesion targeting as well as dosimetrical purposes. In order to deposit the dose to the lesion, tumor was delineated and a dedicated radiation plan was calculated. Therefore, several volumetric modulated arc therapy (VMAT) arcs with an energy of 6 MV were used to deliver the dose of 5 Gy to the lesion while sparing out healthy surrounding tissue. To accurately deliver the calculated dose to the lesion, a cone-beam CT prior to treatment delivery was performed to reposition each mouse separately at the tumor location (Appendix A) [34]. During and after treatment, tumor size and weight were measured with an electrical caliper and tumor volume was calculated using following formula: volume = (length × width^2^) × 0.5.

### 2.15. Statistical Analyses

All analyses were performed using GraphPad Prism 9. Data are expressed as mean ± SEM of at least three independent experiments unless otherwise indicated. Unpaired *t*-test with Mann–Whitney test and one-way ANOVA followed by a Dunnett’s multiple comparison test were used for statistical analyses: * *p* < 0.05, ** *p* < 0.01, *** *p* < 0.001, **** *p* < 0.0001.

## 3. Results

### 3.1. SSP Enzymes Are Expressed in Human Colorectal Cancers

The SSP consists of three enzymatic reactions catalyzed by PHGDH, PSAT1 and PSPH. The serine/glycine hyperactivation signature shows a pathway alteration in 28% of colorectal adenocarcinomas [8]. To assess the clinical relevance of targeting the SSP in CRC for radiosensitizing purposes, we first examined mRNA levels of PHGDH, PSAT1 and PSPH in patient-derived data from normal tissues (TCGA Target GTEx) and CRC tumor tissues (TCGA COAD study) using the online Xena Explorer tool. Data analysis revealed a significant expression gain for PHGDH, PSAT and PSPH in CRC tissues compared to normal colorectal tissues (Figure 1A). Additionally, overall survival (OS) was plotted using patient-derived data from rectum cancer patients and made the distinction between low and high expression of PHGDH, PSAT1 and PSPH. It is evident that a lower OS is correlated with high PHGDH expression, while this is reversed in PSAT1 and PSPH (Figure 1B). However, this should be interpreted with caution, as the number of included patients is relatively low. Altogether, these data indicate CRC as a good model to pursue PHGDH modulation.

Additionally, expression profiles of SSP enzymes in our selected human CRC cell lines, namely HCT116 and DLD-1, were evaluated. Interestingly, our results demonstrated that SSP enzymes are expressed in both cell lines (Figure 1C,D), identifying these as suitable cell lines for further investigation. As aforementioned, PHGDH is the first and rate-limiting enzyme of the SSP, thereby marking PHGDH as an interesting therapeutic target situated at a central metabolic position. Several pharmaceutical inhibitors have been developed against PHGDH [33,35,36], with NCT-503 being the most intensively studied in literature. After establishing nontoxic doses of NCT-503 under normoxic and hypoxic conditions, in absence or presence of exogenous serine (Appendix A), the time frame of inhibition of PHGDH by NCT-503 in both human CRC cell lines was determined by measuring serine production. A first small nonsignificant decrease in levels of intracellular D-serine became apparent after 5 h treatment with NCT-503 in both cell lines (Data not shown).

### 3.2. PHGDH Inhibition Influences Mitochondrial Dynamics and Redox Homeostasis

To evaluate metabolic changes upon PHGDH inhibition, cellular respiration through OXPHOS was assessed. An allosteric feedback loop of serine to pyruvate kinase (PK)M2 exists, which implies that when serine is deprived, PKM2 activity is reduced and glycolysis metabolites are diverted to the SSP, resulting in a reduction in OXPHOS [35]. A drastic dose-dependent reduction in oxygen consumption is revealed, which is already initiated by the lowest used dose of NCT-503 (Figure 2A,B). This reduced cellular respiration is further accompanied with a significant loss of ATP production (Figure 2C) and maximal respiration capacity (Figure 2D). These results indicate a significant change in metabolic activity upon treatment with NCT-503.

The ROS homeostasis is regulated by a balance between oxidants and antioxidants. Alterations in GSH could, among others, potentially lead to a disrupted ROS homeostasis. Hence, we measured the levels of intracellular ROS after exposure to NCT-503 and observed a dose-dependent increase in ROS levels upon treatment with the highest dose of NCT-503 in HCT116 cells (4-fold) (Figure 3D). In line with previous results, ROS levels in DLD-1 cells were significantly increased as well; however, this increase did not occur in a dose-dependent manner and to a lower extent when compared to HCT116 cells (up to 2-fold).

The levels of aKG were additionally measured upon treatment with NCT-503. Notably, aKG is known to improve antioxidative function against oxidative stress in cells. No significant differences in aKG could be measured 5 h after treatment with NCT-503, although we do have to mention that only very limited amounts of aKG could be assessed in every condition, and therefore, these results should be interpreted with caution (Figure 3A). Therefore, we continued our investigation with measuring levels of GSH and its cofactor NADPH. GSH is known to be the most prominent antioxidant present within the cells. Interestingly, both GSH and NADPH are end-metabolites of the SSP. After exposure to NCT-503, GSH levels in HCT116 were decreased in a dose-dependent manner, while no differences in GSH were observed in DLD-1 cells (Figure 3B). Likewise, in HCT116 cells, the levels of NADPH decreased significantly upon treatment with NCT-503, while this decrease was less pronounced in DLD-1 cells (Figure 3C).

### 3.3. Hypoxia Induces the Expression of PHGDH in Human Colorectal Cancer

It has been described that expression of PHGDH is enhanced under hypoxic conditions in TNBC [36]. As hypoxia is a common feature of solid tumors, this could play a role in optimizing therapeutic strategies involving PHGDH inhibition. As mentioned before, hypoxia has been defined as the major contributor to radioresistance and should therefore not be disregarded lightly. Hence, we continued on our online analysis making use of the TCGA database and explored whether a correlation could be observed between hypoxia and PHGDH expression. mRNA-levels of PHGDH were divided amongst low and high (cut-off z-score: 1.5) expression profiles and assessed relative to pre-existing hypoxia scores, namely Winter, Buffa and Ragnum hypoxia scores [37,38,39]. These signatures are based on the differential expression of specific hypoxia-related genes and can be freely accessed through cBioPortal for Cancer Genomics. A significant correlation between high PHGDH expression and hypoxia could be revealed in two out of three predefined hypoxia scores (Winter and Buffa) and a similar trend could be observed in the Ragnum hypoxia score (Figure 4A), indicating that PHGDH expression levels are indeed elevated under hypoxic conditions.

To assess this phenomenon in our culture system, we quantified the protein expression levels of the SSP enzymes in human CRC cell lines HCT116 and DLD-1 in radiobiological-relevant hypoxic conditions (0.1% O_2_) (Figure 4B). An upregulation of all three enzymes could be observed compared to normoxic conditions (Figure 1C,D). The highest fold change between normoxic and hypoxic conditions could be observed for PHGDH in both cell lines, with a particular increase in DLD-1 cells (Figure 4C). The serine production upon treatment with NCT-503 was reassessed in hypoxic conditions (0.1% O_2_) after 5 h of treatment. A similar loss of D-serine production was observed in both cell lines, with a tendency towards dose dependency (Figure 4D). Of note, the overall serine levels in hypoxic conditions convey the impression of being elevated compared to normoxic conditions.

### 3.4. PHGDH Inhibition Disrupts ROS Homeostasis in Human Hypoxic Colorectal Cancer Cells

As assessed under normoxic conditions, we repeated the measurements of relevant intracellular metabolites under hypoxic conditions (0.1% O_2_). Again, aKG levels did not reveal a significant decrease upon treatment with NCT-503 (Figure 5A).

Under hypoxic conditions, the variability between the different experiments increased. As a consequence, no significant decrease in either GSH or NADPH could be observed in both cell lines (Figure 5B,C). However, a similar trend as observed in normoxia was still present, namely a decrease in the levels of GSH in HCT116 cells and a decrease in the levels of NADPH in both cell lines, suggestive for a disrupted ROS balance. Under hypoxic conditions, NCT-503 induced a dose-dependent ROS induction in both cell lines, which already show significance starting from the lowest dose of NCT-503. The highest upregulation of ROS (3-fold, Figure 5D) in HCT116 appears to be lower when compared to normoxia (4-fold, Figure 3D), while in DLD-1 cells, a higher upregulation could be observed (3-fold) compared to normoxia (2-fold). Interestingly, similar to what was observed in the levels of D-serine levels under hypoxic conditions, ROS levels (as measured in MFI) in control cells were significantly higher under hypoxic conditions compared to hypoxia, resulting in a net increase in ROS levels under hypoxia in all conditions (data not shown).

### 3.5. Inhibition of PHGDH Radiosensitizes Human Hypoxic Colorectal Cancer Cells through Excessive ROS

Considering the observed metabolic alterations upon NCT-503 treatment, we investigated whether PHGDH inhibition displayed a radiosensitizing effect in human CRC cells. Experiments were performed both under normoxic (Figure 6A) and hypoxic conditions (Figure 6B,C). Radiosensitization could be observed under hypoxic conditions in both HCT116 and DLD-1 (ER: 1.66 and 1.86, respectively) (Figure 6B), while leaving the intrinsic radiosensitivity unaltered (Figure 6A). Radiosensitizing effects of NCT-503 in DLD-1 were more pronounced, possibly due to the higher upregulation of PHGDH under hypoxic conditions (Figure 4C). Subsequently, we examined whether the observed radio-modulatory effects were the result of the induced ROS production, whereby depleted antioxidants could not rescue the cells from oxidative stress. Using ROS scavenger N-acetyl-cysteine (NAC), the observed radiosensitizing effects from treatment with NCT-503 were completely abolished in both cell lines (Figure 6C) with each used dose of NCT-503 (data not shown). This confirmed the critical role of ROS in radiosensitization through the inhibition of PHGDH.

### 3.6. Inhibition of PHGDH Combined with Radiation Delays Tumor Growth in HCT116 Xenografts

With the promising in vitro data, we examined the combinatorial effect of PHGDH modulation and radiation in HCT116 tumor-bearing mice in a preliminary experiment. We developed an in vivo model in order to locally irradiate mice while preserving their sterile environment. The experiment was set up as follows: HCT116-bearing mice were treated i.p. with NCT-503 at a dose of 40 mg/kg for 4 consecutive days, while fractionated radiation (3 × 5 Gy) was given for 3 consecutive days starting after the second administration of NCT-503. Tumor size was followed up until mice reached any human endpoints (Figure 7A). A fractionated radiation regimen was selected, as this is clinically more relevant compared to a single high dose of radiation. During the optimization experiments, some toxicity was noted after administration of NCT-503. When decreasing the injection volume of NCT-503 from 200 μL to 100 μL (while keeping the dose of NCT-503 at 40 mg/kg), no toxicity was further observed. This led us to hypothesize that the solvent for NCT-503 was responsible for the observed toxicity and not the NCT-503 itself. A decreased tumor growth rate was observed in all treatments compared to control, with a significant delay in reaching a tumor volume of 500 mm3 in all groups (Figure 7B). Of note, mice receiving radiotherapy developed necrotic tumors starting from day 28 after treatment, which ultimately led to these mice being euthanized due to necrotic tumors. Still, survival of the mice was prolonged from a median survival of 28 days in the control group, to 31 days with NCT-503 alone and 37 days with radiotherapy alone. A median survival of 43 days was reached in the combination group, which indicates an additive effect of NCT-503 with fractionated radiation in mice bearing HCT116 tumors (Figure 7C).

## 4. Discussion

Genomic profiling identified an increased metabolic activity of the SSP in a variety of cancer types and this overexpression has been correlated with cancer progression, aggressiveness and treatment resistance [8]. In the present study, we examined the modulation of the SSP by using PHGDH inhibitor NCT-503. Our findings demonstrated that modulation of PHGDH engages oxidative stress in hypoxic conditions mainly through perturbation of the metabolic profile and alteration of the antioxidant system in CRC cells. Furthermore, we have shown that modulation of PHGDH cooperates with radiation in hypoxic CRC through disturbed redox homeostasis and undertook the first steps to evaluate its radiosensitizing potential in HCT116 xenografts. Instead of completely knocking out PHGDH, the use of a PHGDH inhibitor was specifically chosen to safeguard future clinical applicability. It must be mentioned that observations in PHGDH-deficient mice have been made that show abnormalities during embryonic development by malformations in the central nervous system [40]. NCT-503 was designed to engage PHGDH in tumors and displays a selective toxicity, which resembles the toxicity observed in PHGDH knockdown models in vivo; however, off-target effects cannot be completely excluded [33].

Recent studies attempted to determine the importance of PHGDH overexpression and the mechanisms attributed to its expression. In the present study, we established that mRNA levels of SSP enzymes are upregulated in CRC compared to normal tissue, indicating the importance of the SSP in CRC. These results are in line with literature, as it was recently demonstrated that hyperactivation of the enzymes of the SSP occurs in a large variety of distinct cancer types (glioblastoma, esophageal/colorectal adenocarcinoma, (squamous cell) lung/invasive breast carcinoma, melanoma and sarcoma) [8]. Additionally, a correlation was revealed between PHGDH expression and TNM staging in CRC patients, thereby highlighting the possible prognostic role of PHGDH [22]. Furthermore, a correlation between interleukin enhancer-binding factor 3 (ILF3) and reprogramming of the SSP was discovered. ILF3 functions as a critical regulator by increasing SSP gene expression and thereby promoting tumor growth. Of note, ILF3 is frequently overexpressed in CRC patients and correlates with poor prognosis [23]. SSP hyperactivation has been associated with the treatment resistance of chemotherapeutics such as pemetrexed and 5-fluorouracil (5-FU) [8]. Nowadays, 5-FU, as a component of FOLFOX or FOLFIRI, is used as a first-line treatment in CRC in combination with radiotherapy. 5-FU resistance has already been linked to a variety of mechanisms, including the involvement of PSAT1. By removing extracellular serine and knocking out PSAT1, 5-FU resistance could be reversed [41,42]. Taken together, sufficient evidence is available to confirm that overexpression of the SSP plays an essential role in the development of CRC and its response to treatment in the clinic.

Hypoxia-inducible factor (HIF) 1 and HIF2 regulate the expression of distinct SSP enzymes, i.e., SHMT2, and more recently uncovered, PHGDH, PSAT1 and PSPH, indicating a clear link between SSP hyperactivation and hypoxia [8,35]. In agreement, we demonstrated a correlation between overexpression of PHGDH and hypoxia in CRC. Hypoxia is a prominent hallmark of cancer and a well–defined risk factor for clinical resistance to radiation therapy. In CRC, the oxygen consumption rate through OXPHOS is demonstrated to be higher than in normal adjacent tissues. This reverse Warburg phenomenon highlights the feasibility of inhibiting mitochondrial respiration through OXPHOS in CRC regarding radiosensitization. The M2 isoform of pyruvate kinase PK is the predominant isoform in CRC and catalyzes the final step of glycolysis. Low PKM2 activity was demonstrated to drive serine biosynthesis. Importantly, serine and glycine deprivation reduced PKM2 activity in cells, thereby channeling more glucose-derived intermediates into the SSP and altering mitochondrial respiration [35,43]. Therefore, in this study we examined whether inhibition of PHGDH could unsettle the OXPHOS. In line with previous findings [36], we observed an important exhaustion in oxygen consumption upon exposure to NCT-503, which went alongside a significant drop in ATP production and maximal respiration.

To date, radiation therapy is the mainstay treatment for numerous cancer types and induces multiple cellular responses, i.e., apoptosis and cell death through DNA damage as a result of elevated ROS production. Interestingly, end-metabolites of the SSP are tightly involved in the antioxidant system in order to detoxify ROS [28]. It is worthwhile to notice is that aKG is a critical intermediate of the TCA cycle and is simultaneously an intermediate metabolite of the SSP [19]. In line with literature, our findings indicated a clear exhaustion in antioxidant defenses with a decrease in GSH and NADPH levels along with a significant augmentation of ROS production. Contrariwise to what is described in literature, we only observed augmented DNA damage induced by radiation but not by NCT–503 itself (data not shown). Consequently, we further explored the radiosensitizing effect of PHGDH inhibition. Treatment with NCT-503 radiosensitized hypoxic CRC cells at all used doses and observed a tentative additive effect of PHGDH inhibition and radiation in HCT116 xenografts. To date, we have knowledge of one study that investigated the association between the upregulation of serine and radioresistance. This was performed in head and neck squamous cell carcinoma (HNSCC). Radiosensitive and radioresistant HNSCC cell lines from the same donor were compared by liquid chromatography–mass spectrometry and metabolic profiling revealed alterations occurring, among others, in the serine/glycine metabolism after irradiation. The authors hypothesize that the radioresistant cell line could alter its metabolism to control the redox homeostasis, DNA repair and DNA methylation status following irradiation [29].

Cancer cells procure serine and glycine through two distinct approaches: firstly through de novo biosynthesis, and secondly from the environment by membrane transporters such as ASCT2 [44]. In our experimental setup, no additional effects were observed on any of the investigated parameters when omitting extracellular serine (data not shown). Studies have indicated that suppression of PHGDH inhibits cell proliferation in breast cancer in the presence of exogenous serine [45]. However, in vivo, evidence suggests that modulation of the SSP is more effective in delaying tumor growth when mice are fed a serine-free diet [46]. Recently, it was shown that HCT116 and DLD-1, as used in this study, are independent of extracellular serine, while other CRC cell lines such as HT29 and SW48 are dependent on extracellular serine for their growth. This resistance to serine withdrawal has partly contributed to the presence of a KRAS mutation [21]. These observations are in line with our obtained results, where the absence of extracellular serine does not influence any investigated parameter in vitro, suggesting only a negligible role for serine itself related to the observed radiosensitizing effects. Further investigation concerning the influence of extracellular availability of serine and glycine on SSP modulation is imperative to fully understand the underlying mechanisms and bring this therapeutic option to the clinic.

Finally, certain repurposing candidates such as disulfiram (DSF) [47] and regorafenib [48] are currently being investigated as FDA-approved alternatives for their interference with the SSP. DSF, a well-known anti-alcoholic drug, targets the SSP through inhibition of PHGDH [47]. Another potential candidate is regorafenib, commercially known as Stivarga. Regorafenib is a multiple-kinase inhibitor and is indicated for patients with metastatic CRC [49]. It was recently demonstrated that regorafenib stabilized PSAT1, the second enzyme of the SSP pathway. It was further demonstrated that regorafenib caused lethal autophagy by stabilization of PSAT1 in GBM [48]. An additional strategy that merits further exploration is the combinatorial treatment with metformin. Metformin is an antidiabetic drug that exhibits anticancer properties by inhibiting mitochondrial energy metabolism via downregulation of PKM2 [27,50]. Given the allosteric feedback loop of serine to pyruvate kinase (PK)M2, it would be interesting to explore the simultaneous inhibition of PHGDH and PKM2. Furthermore, modulation of PKM2 has already been demonstrated to enhance radiosensitization, which could potentially be further enhanced by the additional modulation of PHGDH. To date, no SSP inhibitors have been developed that can be used in clinic, marking repurposing candidates as promising and exciting future options for serine-targeted therapeutic approaches.

## 5. Conclusions

In conclusion, our results demonstrated that modulation of the de novo serine synthesis pathway, through inhibition of PHGDH, increases radioresponse in hypoxic colorectal cancer cells, while leaving the intrinsic radiosensitivity unaltered. Dysfunction of mitochondrial metabolism and a disrupted ROS homeostasis are, at least partially, responsible for the observed radiosensitizing effects. Further investigation into the dependence of cancer cells on extracellular serine combined with the development of clinical drugs targeting PHGDH will undoubtedly lead to exciting opportunities in the field of cancer research.

## Figures and Tables

**Figure 1 cancers-14-05060-f001:**
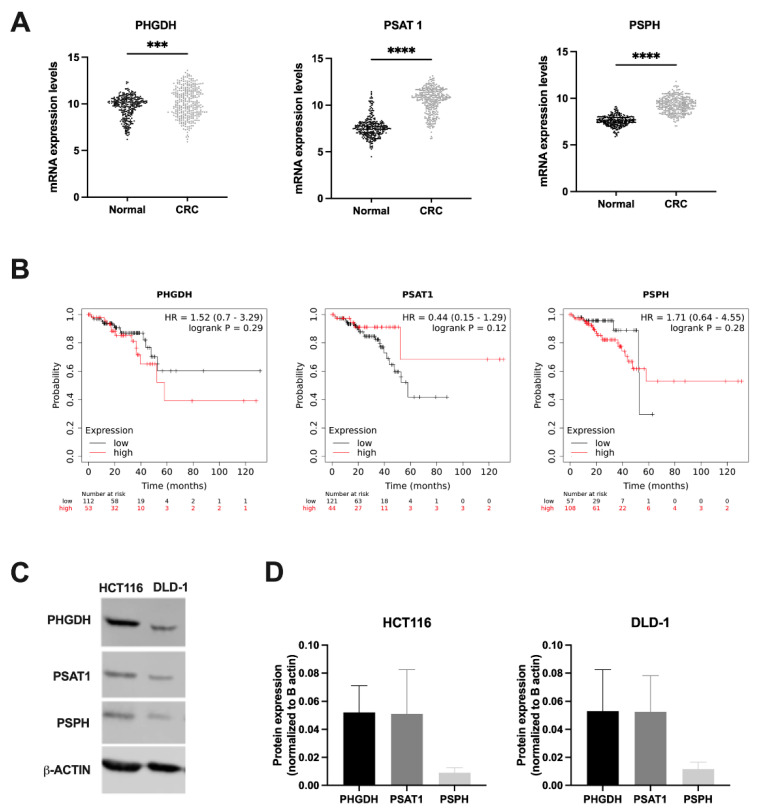
Serine synthesis pathway enzymes are expressed in human colorectal cancers. (**A**) PHGDH, PSAT1 and PSPH mRNA expression levels of GTEx normal tissues were compared to mRNA expression profile of CRC tumor tissues out of the TCGA database (RSEM values). (**B**) Kaplan–Meier curves showing overall survival in PHGDH, PSAT1 and PSPH low/high mRNA-expressing rectum cancer patients out of the TCGA database. (**C**) Representative Western blots of PHGDH, PSAT1 and PSPH with β-actin as loading control in HCT116 and DLD-1 cells under normoxic conditions. (**D**) Relative expression of PHGDH, PSAT1 and PSPH normalized to β-actin in HCT116 and DLD-1 cells under normoxic conditions. Data are shown from at least three replicates as mean ± SEM. Unpaired *t*-test with Mann–Whitney test was used for statistical analysis: *** *p* < 0.001, **** *p* < 0.0001.

**Figure 2 cancers-14-05060-f002:**
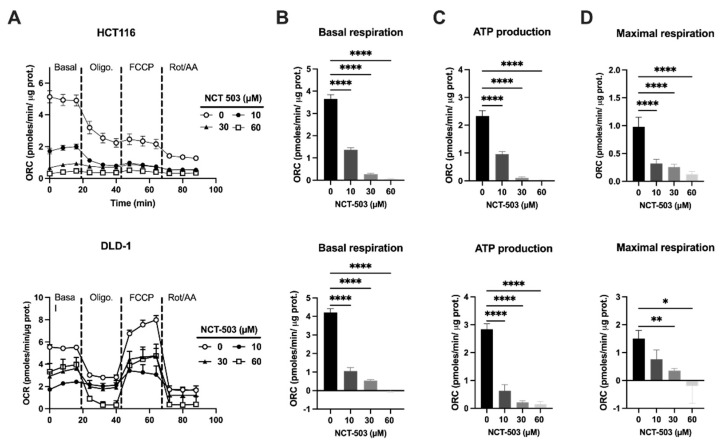
PHGDH inhibition affects mitochondrial cellular respiration under normoxic conditions. (**A**) Representative experiment showing the oxygen consumption rates (OCR) of normoxic HCT116 (top) and DLD-1 (bottom) was measured upon treatment with NCT-503 (16 h) at indicated concentrations after consecutive injection at indicated time points of oligomycin, FCCP, rotenone and antimycin A using the Seahorse analyzer. The OCR was expressed as pmoles/min/µg protein. (**B**) Summary graphs showing basal respiration; (**C**) summary graphs showing ATP production; (**D**) summary graphs showing the levels of maximal respiration. Data are shown from at least three replicates as mean ± SEM. One-way ANOVA with Dunnett’s multiple comparison test was used to calculate statistics: * *p* < 0.05, ** *p* < 0.01, **** *p* < 0.0001.

**Figure 3 cancers-14-05060-f003:**
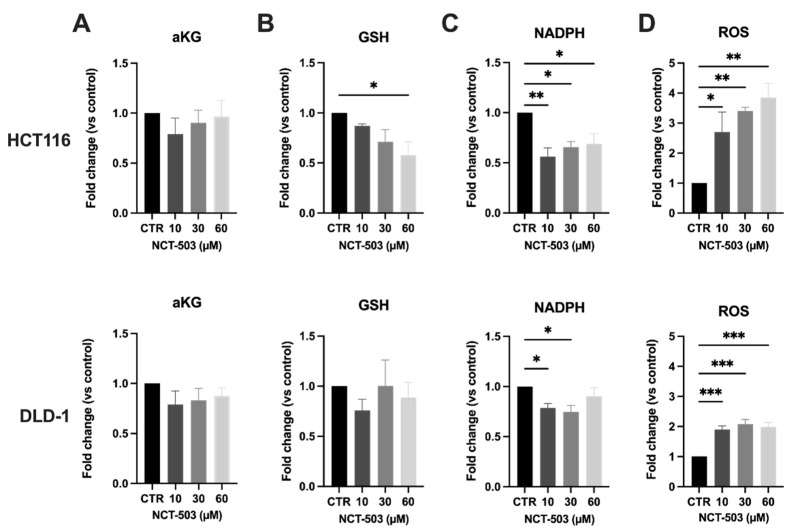
PHGDH inhibition does affect end-metabolites aKG, GSH, NADPH and redox balance in normoxic CRC cells. HCT116 (top) and DLD-1 (bottom) cells were treated with NCT-503 for 5 h at indicated concentrations. (**A**) aKG levels—normalized to control; (**B**) GSH levels—normalized to control; (**C**) NADPH levels—normalized to control; (**D**) ROS levels—normalized to control. Data are shown from at least four replicates as mean ± SEM. One-way ANOVA with Dunnett’s multiple comparison test was used to calculate statistics: * *p* < 0.05, ** *p* < 0.01, *** *p* < 0.001.

**Figure 4 cancers-14-05060-f004:**
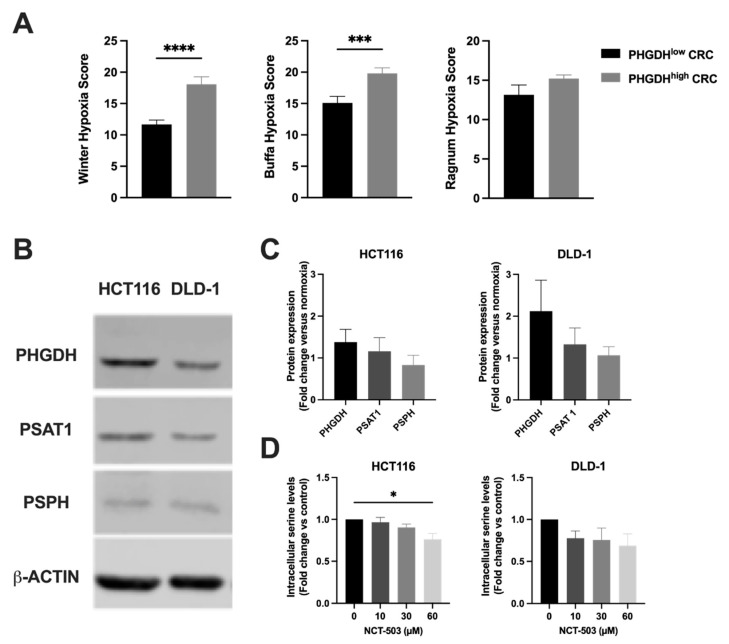
Hypoxia induces the expression of PHGDH in human colorectal cancers. (**A**) Correlation between PHGDH overexpression (*p* ≥ 1.5) and hypoxia by assessing the Buffa, Winter and Ragnum hypoxia scores accessed through cBioPortal for Cancer Genomics. (**B**) Representative Western blots of PHGDH, PSAT1 and PSPH with β-actin as loading control in HCT116 and DLD-1 cells under hypoxic conditions (0.1% O_2_). (**C**) Relative protein expression of PHGDH, PSAT1 and PSPH under hypoxic conditions (0.1% O_2_) normalized to the expression under normoxic conditions. (**D**) D-serine levels were determined upon treatment with NCT-503 (5 h) at indicated concentrations under hypoxic conditions (0.1% O_2_). Unpaired t-test (**A**) and ordinary one-way ANOVA with Dunnett’s multiple comparison test (**D**) were used to calculate statistics: * *p* < 0.05, *** *p* < 0.001, **** *p* < 0.0001.

**Figure 5 cancers-14-05060-f005:**
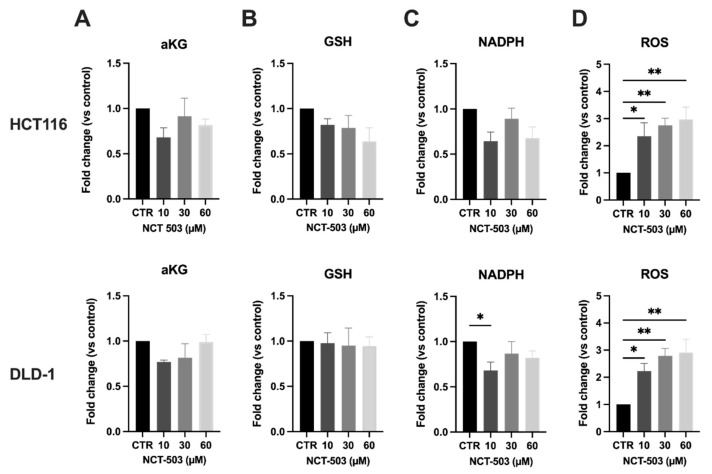
PHGDH inhibition mainly disrupts ROS homeostasis in human hypoxic colorectal cancer cells. HCT116 (top) and DLD-1 (bottom) cells were treated with NCT-503 for 5 h at indicated concentrations under hypoxic conditions (0.1% O_2_). (**A**) aKG levels—normalized to control; (**B**) GSH levels—normalized to control; (**C**) NADPH levels—normalized to control; (**D**) ROS levels—normalized to control. Data are shown from at least four replicates as mean ± SEM. One-way ANOVA with Dunnett’s multiple comparison test was used to calculate statistics: * *p* < 0.05, ** *p* < 0.01.

**Figure 6 cancers-14-05060-f006:**
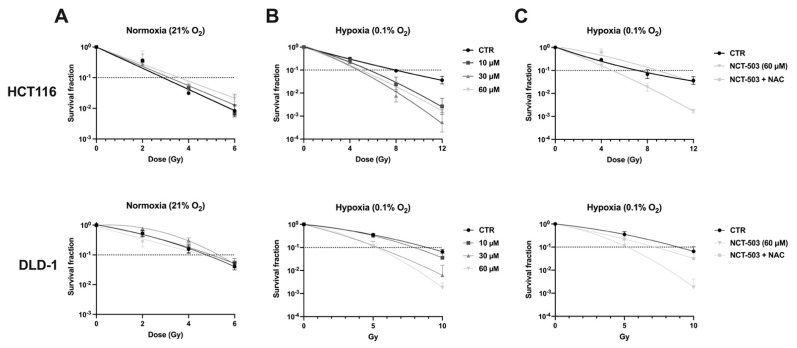
Inhibition of PHGDH radiosensitizes human hypoxic colorectal cancer cells through excessive ROS. HCT116 (top) and DLD-1 (bottom) cells were treated with NCT-503 (16 h) at indicated concentrations, irradiated and reseeded for colony formation assay. (**A**) Dose–response curves under normoxic (**B**) and hypoxic conditions. (**C**) Dose–response curve under hypoxic conditions with treatment of 60 μM NCT-503 with the addition of NAC.

**Figure 7 cancers-14-05060-f007:**
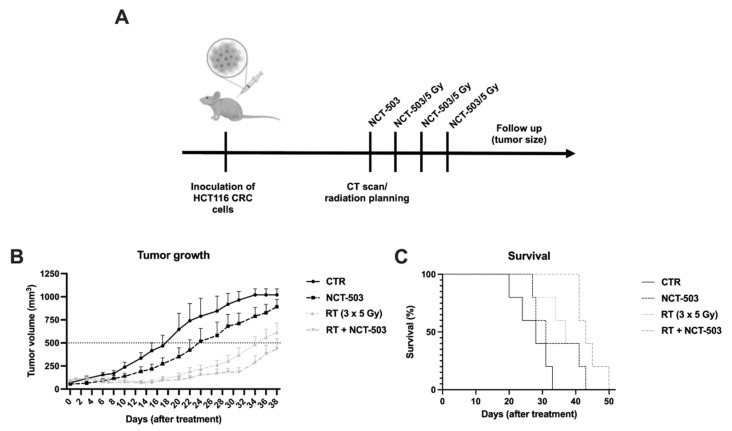
Inhibition of PHGDH combined with fractionated radiation delays tumor growth in HCT116 xenografts. (**A**) Representative schedule of the experimental setup in vivo. (**B**) Tumor growth curve of mice with day 0 as the starting point of treatment. (**C**) Survival curve of mice with day 0 as the starting point of treatment. Data are shown as mean with SEM (*n* = 5).

## Data Availability

Part of the data presented in this study are openly available in TCGA, as discussed in the Material and Methods section.

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
