# Peer review of "Inhibition of Phosphoglycerate Dehydrogenase Radiosensitizes Human Colorectal Cancer Cells under Hypoxic Conditions"

_cancers, 2022, doi:10.3390/cancers14205060_

Round 1
Reviewer 1 Report
In this original article by Van de Gucht et al., the authors examined the modulation of the serine synthesis pathway (SSP) by using the PHGDH inhibitor NCT-503. Their findings demonstrated that modulation of PHGDH engages oxidative stress within hypoxic conditions, mainly through perturbation of the metabolic profile and alteration of the antioxidant system in two CRC cell lines. The modulation of PHGDH was found to cooperate with radiation in hypoxic CRC through disturbed redox homeostasis and undertook the first steps to evaluate its radiosensitizing potential in a xenograft model.
The study is well designed and well presented, the methods are all adequate for reaching the purposes, the results are clear, and the figures and tables all help the understanding of the results.
Their results are discussed in the light of existing knowledge; they are partially integrated into existing ideas; and on the other hand, they open up new directions for further research.
I suggest accepting the manuscript for publication.
Reviewer 2 Report
The authors have explored the potential of targeting de novo serine biosynthesis to radio-sensitize the hypoxic colorectal cancer cells using PHGDH inhibitor NCT-503. The authors have used several biochemical assays to ascertain the findings.
Comments:
1. The prognostic implications of PHGDH, PSAT1, and PSPH may be analyzed in publicly available datasets, e.g., TCGA to strengthen the data. A similar analysis may be performed to analyze the prognostic significance of “PHGDH expression levels are indeed elevated 358 under hypoxic conditions.”
2. For further validation of ROS generation and Apoptosis, the authors may use immunoblot analysis of several well-established protein markers.
3. The authors may justify the selection of CRC cell lines with corresponding references.
4. The authors may ensure that the ‘metabolic changes upon PHGDH inhibition’ are not off-target effects of NCT-503 with proper controls. Here, the authors may validate using genetic knock-down/knock-out models may be required to establish the findings. The authors may also discuss it and include it in the Discussion section.
5. The discussion may be revised to highlight all the obtained results.
Reviewer 3 Report
The study by Van De Gucht et al has explored the effect of phosphoglycerate dehydrogenase (PHGDH) inhibition on the radioresponse in colorectal cancer (CRC) cell lines and in an in vivo mouse model under aerobic and radiobiological relevant hypoxic conditions. Using a multi-pronged approach such as analysis of patient-derived mRNA expression profiles and biochemical analysis of CRC cell lines, the authors found that the levels of the key enzymes involved in serine synthesis pathway (SSP) are upregulated in CRC compared to normal tissue. The overexpression of PHGDH correlated with hypoxia in CRC. The authors uncovered potential metabolic changes that drove the increased radioresponse under hypoxic conditions upon PHGDH inhibition. Oxygen consumption, ATP production and maximal respiration dropped upon PHGDH inhibition, which was indicative of impaired cellular respiration through OXPHOS. Concomitantly, increased ROS production occurred due to decreased levels of GSH and NADPH. Finally, the authors showed that inhibiting PHGDH in combination with fractionated radiation delayed tumor growth in HCT116 xenografts.
Overall, the study is well carried out and easy to follow. The manuscript is clearly written. The rationale for each experiment design is clearly laid out. The discussion is very thorough. My only major concern is that data for figure 5 is not what the text describes. The result section for figure 5 describes measurements of relevant intracellular metabolites under hypoxic conditions. However, figure 5 shows cell viability under normoxia and hypoxia conditions in the presence and absence of exogenous serine. Please rectify. Some typographical errors are present in the manuscript, for example line 606: SPP.
